



# Differentiation of coarse-mode anthropogenic, marine and dust particles in the high Arctic Islands of Svalbard

Congbo Song[1], Manuel Dall'Osto[2], Angelo Lupi[3], Mauro Mazzola[3], Rita Traversi[4,5], Silvia Becagli[4,5], Stefania Gilardoni[3], Stergios Vratolis[6], Karl Espen Yttri[7], David C. S. Beddows[1], Julia Schmale[8], James Brean[1], Agung Ghani Kramawijaya[1], Roy M. Harrison[1,a], and Zongbo Shi[1]

[1]School of Geography Earth and Environment Sciences, University of Birmingham, Birmingham B15 2TT, UK
[2]Institute of Marine Science, Consejo Superior de Investigaciones Científicas (CSIC), Barcelona, Spain
[3]Institute of Polar Sciences, National Research Council (CNR-ISP), 40129 Bologna, Italy
[4]Department of Chemistry "Ugo Schiff", University of Florence, Via della Lastruccia 3, 50019 Sesto Fiorentino, Italy
[5]Institute of Polar Sciences, ISP-CNR, Via Torino 155, 30172 Venice-Mestre, Italy
[6]ERL, Institute of Nuclear & Radiological Sciences & Technology, Energy & Safety, National Centre of Scientific Research Demokritos, 15310 Ag. Paraskevi, Attiki, Greece
[7]NILU -Norwegian Institute for Air Research, P.O. Box 100, N-2027 Kjeller, Norway Kjeller, Norway
[8]School of Architecture, Civil and Environmental Engineering, École Polytechnique Fédéderale de Lausanne, Lausanne, Switzerland
[a]also at: Department of Environmental Sciences/Center of Excellence in Environmental Studies, King Abdulaziz University, P.O. Box 80203, Jeddah, 21589, Saudi Arabia

**Correspondence:** Manuel Dall'Osto (dallosto@icm.csic.es) and Zongbo Shi (z.shi@bham.ac.uk)

**Abstract.** Understanding aerosol-cloud-climate interactions in the Arctic is key to predict the climate in this rapidly changing region. Whilst many studies have focused on submicron aerosol (diameter less than 1 $\mu$m), relatively little is known about the climate relevance of supermicron aerosol (diameter above 1 $\mu$m). Here, we present a cluster analysis of multiyear (2015-2019) aerodynamic volume size distributions with diameter ranging from 0.5 to 20 $\mu$m measured continuously at the Gruvebadet

Observatory in the Svalbard archipelago. Together with aerosol chemical composition data from several online and offline measurements, we apportioned the occurrence of the coarse-mode aerosols to anthropogenic (two sources, 27%) and natural (three sources, 73%) origins. Specifically, two clusters are related to *Arctic haze* with high levels of black carbon, sulfate and accumulation mode (0.1-1 $\mu$m) aerosol. The first cluster (9%) is attributed to ammonium sulfate-rich *Arctic haze* particles, whereas the second one (18%) to larger-mode aerosol mixed with sea salt. The three natural aerosol clusters were: open ocean

sea spray aerosol (34%), mineral dust (7%), and an unidentified source of sea spray-related aerosol (32%). The results suggest that sea spray-related aerosol in polar regions may be more complex than previously thought due to short/long-distance origins and mixtures with *Arctic haze*, biogenic and likely snow-blowing aerosols. Studying supermicron natural aerosol in the Arctic is imperative for understanding the impacts of changing natural processes on Arctic aerosol.



## 1    Introduction

The Arctic is one of the most sensitive regions of the world, and the Arctic environment is experiencing tremendous changes at a much faster pace than lower latitudes (Landrum and Holland, 2020). The rising temperature, sea ice melt and local air pollutant emissions (Schmale et al., 2018) in the Arctic all exert a broad range of impacts on natural and anthropogenic processes, thereby changing Arctic aerosol properties and consequently radiative forcing and cloud formation (Abbatt et al., 2019; Willis et al., 2018). A better knowledge of Arctic aerosol is an essential requisite for narrowing the uncertainty in assessing the impacts

of aerosols on cloud formation and climate change. It is known that aerosol-cloud-climate interactions depend upon aerosol properties, such as concentration, size distribution and chemical composition. In particular, the size distribution of aerosols is important as size dictates many of the direct and indirect climate forcing properties of aerosols, as well as indicating their sources and atmospheric lifetimes.

There are several studies that have investigated particle number size distributions at high Arctic sites (Dall'Osto et al., 2019;

Freud et al., 2017; Asmi et al., 2016), but the spectra generally measured by Scanning Mobility Particle Sizers (SMPS) are usually limited to aerosol diameters of less than 1 $\mu$m, i.e., submicron aerosol. These studies are highly informative, but the measured submicron aerosols are unlikely to capture some important aerosol sources that are mainly present in the supermicron mode (i.e., above 1 $\mu$m diameter) decisive for the volume/mass concentration basis, notably sea spray and mineral dust (Quinn et al., 2015; Ricard et al., 2002; Porter and Clarke, 1997; Li and Winchester, 1990). Recently, a long-term trend analysis

demonstrated that aerosol observed at the Zeppelin Observatory has become more dominated by coarse mode over time, most likely due the result of increases in the relative amount of sea spray aerosol (Heslin-Rees et al., 2020).

Sea spray aerosol (SSA) is primarily generated from the wave breaking process resulting in bubble bursting at the sea surface, and is a significant contributor to the Arctic aerosol mass burden (Quinn et al., 2002; Fitzgerald, 1991). SSA originate from film drops (fragment of the collapsed cap), jet drops (shot from the cavity formed after the bubble bursting) and spume drops

(torn of the wave crest by sufficient high wind stress) (Feng et al., 2017; Quinn et al., 2015). The film drops are evaporated in the atmosphere and generate submicron SSA while the jet and spume drops can produce supermicron SSA (O'Dowd et al., 1997). According to size-resolved chemical composition of individual SSA reproduced in a laboratory facility, submicron SSA is overwhelmingly dominated by organic matter and sea salt mixed with organic matter while supermicron SSA mainly consists of sea salt and biological species (Prather et al., 2013). This suggests that understanding supermicron aerosol in the Arctic is of

great value for understanding not only sea salt but also biological processes. Contrary to other global oceans, in the Arctic sea salt can be generated from open water, open leads in the sea ice (Kirpes et al., 2019; May et al., 2016) and salty blowing snow (Frey et al., 2020; Huang and Jaeglé, 2017). All sources can contribute significantly to the Arctic sea salt budget and they have different seasonality. With rapidly declining Arctic sea ice extent and snow coverage while increasing ocean areas, the chemical nature and size distribution of SSA in the Arctic are likely to change, which potentially alters cloud condensation nuclei (CCN)

activity (Collins et al., 2013), ice nucleating capacity (Wilson et al., 2015; Russell, 2015) and radiative forcing (Struthers et al., 2011). Murphy et al. (2019) recently also reported a source of sea-salt aerosol over pack ice that is distinct from that over open water, stressing that sea spray aerosols in the Arctic regions may be more complex than previously assumed.



Mineral dust is important owing to its ability to affect the radiation balance of the atmosphere as well as the surface energy balance by its presence in layers over high albedo surfaces or deposited on snow and ice (Kylling et al., 2018; Stone et al.,

2007). They can also serve as ice nucleating particles and thus altering the cloud properties and lifetime (Tobo et al., 2019). Sharma et al. (2019) found dust components in the Arctic to be most abundant in late summer/early autumn, whereas a less abundant but still significant peak was observed in spring. The atmospheric dust load in the Arctic is presumably influenced by high-latitude local dust emissions (Tobo et al., 2019; Bullard et al., 2016) and long-range transported windblown dust from low-mid latitudes (Sirois and Barrie, 1999). Mineral species in Arctic aerosol were reported to be mainly present in the

supermicron mode with mean aerodynamic diameter at ∼5 $\mu$m on a volume/mass concentration basis (Ricard et al., 2002). The frequency of windblown dust events is projected to increase in the Arctic as a consequence of the rapid and widespread retreat of glaciers leading to more ice free terrain (proglacial fields or floodplains where fine glaciofluvial sediment deposits are exposed to wind) (Tobo et al., 2019; Bullard et al., 2016).

Measuring supermicron aerosol in the Arctic is important to understand the chemical nature of SSA and mineral dust, their

potential sources as well as their impacts on aerosol-radiation and aerosol-cloud-interactions in the rapidly changing Arctic. Several studies have measured supermicron aerosol in the Arctic environment using multi-stage cascade impactors combined with offline chemical analysis (Ghahremaninezhad et al., 2016; May et al., 2016; Ricard et al., 2002; Li and Winchester, 1990), and by single-particle analysis (Tobo et al., 2019; Kirpes et al., 2019; Yu et al., 2019; Kirpes et al., 2018; Chi et al., 2015; Geng et al., 2010). These measurements were typically carried out for short periods in a particular season, serving specific research

questions, providing limited understanding of the seasonal cycle of the supermicron aerosol in the Arctic. One exception is the multiyear (2006-2009) measurements of submicron and supermicron aerosols by May et al. (2016), who found that open leads contribute to local Arctic SSA emissions year-round. The aerosol size distribution, seasonal, and interannual variations of the supermicron aerosol remain open questions due to the low time resolution as well as the limited aerosol size bins.

To improve current understanding of the seasonal cycle and interannual variations of the submicron and supermicron

aerosols, as well as the roles of natural and anthropogenic aerosols in the Arctic, in situ aerosol aerodynamic size distributions up to 20 $\mu$m were measured from 2015 to 2019 at an hourly time resolution at a background site in Svalbard, i.e Gruvebadet. A $k$-means clustering technique was applied to the size distribution to separate major aerosol types (Lachlan-Cope et al., 2020; Dall'Osto et al., 2019; Freud et al., 2017; Beddows et al., 2014, 2009). In addition, the aerosol size distributions are complemented by aerosol chemical composition data from several online/offline measurements to distinguish anthropogenic

and natural aerosol types as well as their potential sources. This study adds to our knowledge of supermicron aerosol in the rapidly changing Arctic, and it is a complement to the submicron aerosol number size distributions in the European Arctic presented by Dall'Osto et al. (2019). Future studies will look at receptor modelling of both aerosol size distributions and chemical composition in the Arctic.



## 2 Materials and Methods

### 2.1 Sampling and measurements

Five years (from 2015 to 2019) of aerosol sampling and measurements were carried out at the Gruvebadet (GVB) Observatory (78.918°N, 11.895°E; 61 m above mean sea level), located at about 800 m south-west from the village of Ny-Ålesund in the Svalbard archipelago. In the north-east direction towards the Ny-Ålesund research village, a clean area was established and motorized activity and other potentially contaminant activities were forbidden. The geographic location of GVB and the

dominant winds ensure a minimal anthropogenic contamination from local emissions (Udisti et al., 2016), whilst being able to capture long-range transported pollution air masses.

The aerosol size distribution was measured by an Aerodynamic Particle Sizer (APS; TSI model 3321, 52 channels) (Traversi et al., 2020). The APS spectrometer provides real-time aerodynamic measurements of particles from 0.5 to 20 $\mu$m diameter. Typical single-channel uncertainty in aerosol number concentration measured by the APS is $\pm$10% according to the specification

sheets. The aerosol size distribution was usually measured from March to September (see Fig. 1), with approximately 21200 hourly observations in total from 2015 to 2019, corresponding to approximately 909 and 140 aerosol size distributions after averaging to daily and weekly observations, respectively. Aerosol light absorption was simultaneously measured at three wavelengths (i.e., 467 nm, 530 nm and 660 nm) by a Particle Soot Absoprtion Photometer (PSAP, Radiance Research) at a 1-minute time resolution and then averaged into hourly observations. The raw data from PASP were corrected according to

Virkkula et al. (2005). Aerosol absorption coefficient is directly proportional to the concentration of equivalent black carbon (eBC) (Petzold et al., 2013), a marker of anthropogenic and wildfire emissions in the Arctic (Abbatt et al., 2019). Details regarding the PSAP and eBC measurements at the site are described by Gilardoni et al. (2020). The APS and PSAP are attached to a same inlet, which follows EUSAAR-ACTRIS protocol since 2011 and runs about 4 m above the ground. Further information can be found elsewhere (Lupi et al., 2016).

Aerosol filter samples were collected daily from February 2015 to August 2018, using a Tecora SkyPost sequential sampler equipped with a PM$_{10}$ sampling head, operating following the EN 12341 European protocol. PM$_{10}$ was collected on Teflon filters (Pall Corp., 47 mm diameter, 2 $\mu$m porosity, collection efficiency 99.6%); 606 daily samples were analysed of which 570 samples overlapped with the aerosol size distribution measurements. Samples were handled in a clean room (class 10,000), under a laminar flow hood (class 100) to minimize contamination. One half of each filter sample was extracted in an ultrasonic

bath (15 min) using ultrapure water (MilliQ, 18 M$\Omega$ cm, 12 mL) for subsequent analysis of inorganic anions (Cl$^-$, Br$^-$, NO$_3^-$ and SO$_4^{2-}$) and cations (Na$^+$, NH$_4^+$, K$^+$, Mg$^{2+}$ and Ca$^{2+}$), and selected organic anions (methanesulfonate (MSA) and oxalate) on a three Dionex ion chromatography (IC) system, equipped with electrochemical-suppressed conductivity detectors. The configuration of the three IC-system is described by Giardi et al. (2016) and Udisti et al. (2016, 2012). Analytic uncertainty is typically below 5%.

Meteorological parameters, including wind speed, wind direction, relative humidity and ambient temperature, were recorded hourly by a Vaisala thermo-hygrometer model HMP45AC and a Young Marine wind sensor model Wind Monitor 05106 at



the height of 10 m (a.g.l.) on the Amundsen-Nobile Climate Change Tower (Mazzola et al., 2016) in the neighborhood of Gruvebadet observatory.

## 2.2  $k$-means cluster analysis

Cluster analysis is commonly used to interpret particle number size distribution measurements (Lachlan-Cope et al., 2020; Dall'Osto et al., 2019; Freud et al., 2017; Beddows et al., 2014, 2009). As we mainly focus on supermicron aerosol, we conduct cluster analysis of the aerosol volume size distribution rather than aerosol number size distribution using a $k$-means clustering algorithm following a standard procedure by Beddows et al. (2014). The underlying principle of the $k$-means clustering is to minimize the sum of squared Euclidean distances between each data point and the corresponding cluster centres (the average spectra of the subgroup). This allows for dividing the whole dataset into a predefined number of subgroups, which are as different as possible from each other, and as coincident as possible within themselves.

To cluster the spectra modes irrespective of the magnitude of the concentrations, the aerosol size distributions were pre-processed using min-max normalization:

$$y_i = \frac{x_i - \min_{1 \leq j \leq n}\{x_j\}}{\max_{1 \leq j \leq n}\{x_j\} - \min_{1 \leq j \leq n}\{x_j\}} \tag{1}$$

where $x$ and $y$ are the absolute and normalized concentrations, respectively. The subscript $i$ and $j$ denoted size bins. $n$ is the number of size bins. The normalized concentrations were then clustered using $k$-means.

## 2.3  Source apportionment of sulfate

Source apportionment of sulfate was carried out following Udisti et al. (2016) to understand potential sources of sulfate for the identified aerosol clusters. We first apportion $Na^+$ and $Ca^{2+}$ to sea salt and non-sea salt fractions. In general, $Na^+$ and $Ca^{2+}$ are considered as proxies for sea salt and mineral dust, respectively. However, a small amount of $Na^+$-containing aerosol in remote regions could be brought in by non-sea salt sources and some fractions of $Ca^{2+}$-containing aerosol originate from sea spray (Salter et al., 2016; Bigler et al., 2006). Assuming a $Ca^{2+}/Na^+$ ratio (w/w) of 0.038 for sea water and 1.78 for mineral dust (Bowen et al., 1979), the measured total $Na^+$ and $Ca^{2+}$ can be apportioned to a sea salt (ss) fraction and a non-sea salt (shortened to nss) fraction, calculated by:

$$ss\text{-}Na^+ = Na^+ - nss\text{-}Ca^{2+}/1.78 \tag{2}$$

$$nss\text{-}Ca^{2+} = Ca^{2+} - 0.038 \times ss\text{-}Na^+ \tag{3}$$

The sea-salt fraction of sulfate (ss-$SO_4^{2-}$) was calculated by multiplying the ss-$Na^+$ (as a sea spray marker) concentration by 0.253 (indicating the $SO_4^{2-}/Na^+$ w/w ratio in sea water) (Bowen et al., 1979). The non-sea salt fraction of sulfate (nss-$SO_4^{2-}$)



was calculated by subtracting the ss-$SO_4^{2-}$ contribution from the total $SO_4^{2-}$ concentrations. The mineral fraction of sulfate (mineral-$SO_4^{2-}$) was estimated by multiplying the nss-$Ca^{2+}$ content by 0.59 ($SO_4^{2-}$/$Ca^{2+}$ w/w ratio in uppermost earth's crust) (Guerzoni and Chester, 1996). Sulfate formed by atmospheric oxidation of dimethylsulfide (DMS) from micro-algae, bio-$SO_4^{2-}$, was estimated by multiplying methanesulfonate (MSA) concentration by 3.0, which was derived from the relationship between MSA and $SO_4^{2-}$ at the same site by Udisti et al. (2016). In their study, they used 136 $PM_{10}$ samples from the

spring-summer season of 2014 to establish the MSA and bio-$SO_4^{2-}$ relationship. The anthropogenic fraction of sulfate (anthr-$SO_4^{2-}$) in each sample is then estimated by subtracting the sum of the ss-$SO_4^{2-}$, mineral-$SO_4^{2-}$, and bio-$SO_4^{2-}$ fractions from the total $SO_4^{2-}$ concentration.

### 2.4    Back trajectories and Potential source regions

The Hybrid Single Particle Lagrangian Integrated Trajectory (HYSPLIT 5.0.0) model (Stein et al., 2016) was used to calculate
hourly backward trajectories arriving at an altitude of 100 m (average mean sea level) at the GVB from 2015 to 2019. The determination of the length of the back trajectories is a compromise between the uncertainty in calculation, which increases with time, and the typical lifetime of aerosol in the Arctic, which is usually between 3 and 8 days for black carbon (Stohl, 2006) and up to 39 days during *Arctic haze* (Baskaran and E. Shaw, 2001). In the present study, a 7-day air mass backward trajectory was chosen.

The potential source contribution function (PSCF) values were calculated using the "openair" package in R (Carslaw and Ropkins, 2012) to identify the potential geographic origins of each aerosol cluster. The geographic region covered by the back trajectories was divided into $1°\times1°$ grid. The PSCF value is calculated by:

$$PSCF_{i,j} = (m_{i,j}/n_{i,j})W_{i,j} \tag{4}$$

where $n_{i,j}$ is the total number of endpoints that fall in the $i,j$th cell, and $m_{i,j}$ is defined as the number of endpoints in the same
cell that exceed the threshold criterion. In the present study, the $75^{th}$ percentile of total volume concentration for each cluster was used as the criterion value. A higher ratio of $m_{i,j}/n_{i,j}$ indicates a higher probability of a particular grid through which a passing air parcel would result in a higher concentration at the receptor (Zeng and Hopke, 1989). $W_{i,j}$ is an empirical weight function to reduce the uncertainty of $m_{i,j}/n_{i,j}$ for the cells with $n_{i,j}$ less than 2-times the grid average number of the trajectory endpoint ($2\times n_{ave}$):

$$W_{i,j} = \begin{cases} 1.0, & n_{i,j} > 2n_{ave} \\ 0.75, & n_{ave} < n_{i,j} \leq 2n_{ave} \\ 0.5, & 0.5n_{ave} < n_{i,j} \leq n_{ave} \\ 0.15, & 0 < n_{i,j} \leq 0.5n_{ave} \end{cases} \tag{5}$$





## 3   Results

Figs. 1 and A1 show the monthly average aerosol volume and number size distributions from 2015 to 2019, respectively. The peaks of monthly average number size distributions all occurred in the accumulation mode and the number concentration (dN/dlogDp) of the peaks are usually below 10 cm$^{-3}$ (Fig. A1). For volume size distribution, bimodal or trimodal size
distributions are often observed throughout the years. The highest concentration was found at 2-4 $\mu$m diameter, regardless of season. A second peak was observed in the submicron mode at 0.6-0.8 $\mu$m diameter, more prominent from March to May.

### 3.1   Characterization of the $k$-means derived aerosol types

The $k$-means cluster analysis of aerosol volume size distributions was performed using 909 daily spectra collected from 2015 to 2019. The determination of the most appropriate number of clusters is critical. Here, the Dunn Index (DI) and Silhouette
Width (SW) were calculated for cluster numbers ranging from 2 to 15 (see Fig. A2). DI is to identify sets of clusters that are compact, with a small variance between members of the cluster, and well separated, where the cluster centres are sufficiently far apart, as compared to the within cluster variance. A higher DI indicates better separated clustered. SW refers to a method of interpretation and validation of consistency within clusters of data. A high SW indicates that an individual spectrum is well matched to its own cluster and poorly matched to neighbouring clusters. For a given assignment of clusters, higher DI
and SW values indicate better clustering results. According to the DI and SW values versus cluster numbers (Fig. A2), five- and seven-cluster solutions appear to be the two promising solutions. For the seven-cluster solution, we observed an obvious increase in the DI value but a decrease in the SW value compared to those from the five-cluster solution, suggesting the some of the clusters in the seven-cluster solution tend to be similar to neighbouring clusters. Fig. A3 illustrates that the C3 to C6 aerosol categories overlapped with each other and are not well separated, especially when taking into account the standard
deviation. The five-cluster solution is able to overcome the overlapping issues and thus was chosen in the present study.

Note that the five clusters (C1, C2, C3, C4 and C5) were ranked by the main aerodynamic diameter of the peak concentration in the average spectrum, from small to large diameters for C1 to C5 (Fig. 2). Fig. 3 shows the daily temporal trends and the polar plots of the five aerosol clusters. The aerosol types are characterized as follows:

- C1: Occurring 9.3±5.1% ($\mu \pm \sigma$, average from monthly occurrence frequency) of the time. C1 presents a bimodal size
distribution with a dominant peak in the accumulation mode at ∼0.7 $\mu$m and a lower peak at ∼3 $\mu$m (2a-b and 3a). The predominant months were March (15.9%) and April (18.8%), i.e., in spring (Fig. 2c).

- C2: Occurring 17.6±17.4% of the time. C2 shows a bimodal size distribution with peaks at ∼0.9 and 2-3 $\mu$m (2a-b and 3a). The concentration in the accumulation mode was lower but with a slightly larger diameter (∼1 $\mu$m) compared to C1. The predominant months are from January (43.2%) to March (32.2%), i.e., in winter and early spring (Fig. 2c).

- C3: Occurring 34.2±8.2% of the time. The size distribution is dominated by the coarse mode with a peak at ∼3 $\mu$m (2a-b and 3a). C3 occurred throughout all seasons, with a relative contribution ranging from 19.2% in October to 49.3% in November (Fig. 2c).





- C4: Occurring 32.0±18.7% of the time. The size distribution is dominated by the coarse mode, with a peak diameter
  of 3-4 $\mu$m (2a-b and 3a). The average peak concentration for C4 is slightly higher and with a slightly larger diameter
  compared to C3. This cluster dominated over other clusters from June to November (Fig. 2c).

- C5: Occurring 6.9±5.5% of the time. C5 shows a trimodal distribution with a peak in the accumulation mode at ∼0.7
  $\mu$m, and in the coarse mode at 3-4 $\mu$m and at 12-14 $\mu$m (2a-b and 3a). The predominant months are from June (8.9%) to
  October (15.0%) with warm temperature and low relative humidity (Fig. A4), i.e., in summer and early autumn.

The occurrence of the five clusters and the key chemical species associated with each cluster are summarized in Table 1.
The interannual variations of fraction of occurrence from March to June are also shown in Fig. 2d-g. No clear interannual
patterns were observed for the aerosol clusters identified. The polar plots show that high concentrations of the five clusters
were typically found when wind speed was higher than 5 m s$^{-1}$ (Fig. 3a), indicating limited impact of local emissions on
coase-mode particles at the GVB station except for wind lifted particles.

## 3.2 Relationship of aerosol clusters and chemical components

Fig. 4 and Fig. 5 present average concentrations of each chemical species and absorption coefficients for the identified clusters,
respectively. Absorption coefficients (Fig. 5b), oxalate (Fig. 4j), total $SO_4^{2-}$ (Fig. 4h) and $NH_4^+$ (Fig. 4f) were enhanced for C1.
These components are also enriched in C2, along with $Na^+$ (Fig. 4d), $Cl^-$ (Fig. 4e) and $Br^-$ (Fig. 4i). Absorption coefficients
were relatively low for C3-5 compared to C1 and C2 (see Fig. 5b). MSA (Fig. 4k), $Na^+$ (Fig. 4d) and $Cl^-$ (Fig. 4e) were
enriched in C3. C4 showed an enhanced concentration in MSA (Fig. 4k), $Na^+$ (Fig. 4d) and $Cl^-$ (Fig. 4e), whereas C5 was
characterized by lowest $K^+$ (Fig. 4a), $NH4_4^+$ (Fig. 4f), $NO_3^-$ (Fig. 4g) and $SO_4^{2-}$ (Fig. 4h). The relationship between C1-5 and
the absorption coefficient is shown in Fig. 5b. Clearly, absorption coefficient for C1-2 was enhanced in comparison to that for
C3-5. We will discuss these results in detail in section 4.

$Na^+$ and $Ca^{2+}$ are usually apportioned to a sea salt and a non-sea salt fraction using characteristic ratios of $Ca^{2+}$ and $Na^+$
in sea water and in the Earth's crust (Eq. 2-3). Fig. 6 shows a scatter plot of $Ca^{2+}$ versus $Na^+$ for the clusters, including
$Ca^{2+}/Na^+$ ratios for sea water (0.038, w/w) and the Earth's crust (1.78, w/w). The regression line of $Na^+$ and $Ca^{2+}$ for C5 is
close to the "mineral" ratio whereas it is closer to the sea water ratio for C2-4. Note that the $Na^+$ and $Ca^{2+}$ regression line for
C3 fits very well with that in sea water. The regression line for C1 is in the middle of the mineral ratio and the sea water ratio.

The apportioned ss-$SO_4^{2-}$, mineral-$SO_4^{2-}$, bio-$SO_4^{2-}$, anthr-$SO_4^{2-}$, and sea salt aerosol (Fig. 7) further improves our understanding
of the aerosol clusters-associated sources. C2-4 was enhanced with respect to ss-$SO_4^{2-}$, ranging from 66.1 ng m$^{-3}$ in C2 to
78.9 ng m$^{-3}$ in C4, as well as SSA (ranging from 853.4 ng m$^{-3}$ in C2 to 1019.2 ng m$^{-3}$ in C4). A high bio-$SO_4^{2-}$ fraction
was observed in C3 (86.2 ng m$^{-3}$) and C4 (61.6 ng m$^{-3}$), whereas mineral-$SO_4^{2-}$ are elevated in C1 (7.7 ng m$^{-3}$) and C5 (7.9
ng m$^{-3}$). The anthr-$SO_4^{2-}$ level decreased from C1 (763.8 ng m$^{-3}$) to C5 (49.7 ng m$^{-3}$). Anthr-$SO_4^{2-}$ contributes to 34.4% in
C5 to 91.7% in C1.





### 3.3 Potential source-areas of the aerosol clusters

The PSCF probability map (Fig. 8) for C1 illustrates that air mass origin was mainly from North-eastern Russia (lat>60°). Potential geographical source regions of C2 are more complex and likely distributed in northern Eurasia (lat>60°), western and northern coast of Greenland, the Norwegian Sea, the Barents Sea and the North Sea (lat>55°). In contrast to C1-2, several regions of elevated probability for C3 appear in open sea/ocean regions (lat>55°), including the Norwegian Sea, the North Atlantic Ocean, the Barents Sea, the Kara Sea and the Baffin Bay. C4 is likely associated to both land/snow (e.g., Greenland,

Northern Europe, north-western Russia) and sea/ocean regions (e.g., the Greenland Sea, the Norwegian Sea, the North Atlantic Ocean, the Barents Sea and the Baffin Bay). C5 exhibits high PSCF values near the western/northern coast of Greenland (e.g., Kangerlussuaq area), northern Eurasia and northern Alaska. For the whole aerosol population, high PSCF values were broadly observed in the surrounding sea/ocean regions and occasionally observed in land/snow regions (e.g., northern Eurasia and Greenland).

## 4 Discussion

### 4.1 Identification of potential sources of the aerosol clusters

The potential sources of the aerosol clusters were identified based on (1) volume size distribution, (2) chemical components, (3) seasonality in occurrence of the clusters and (4) potential source maps.

#### 4.1.1 C1: Anthropogenic *Arctic haze*

The distinct peaks of C1 are typical modes of *Arctic haze* (Radke et al., 1984). The high occurrence of C1 in March and April are consistent with typical months of springtime *Arctic haze* (Abbatt et al., 2019), which has the highest absorption coefficient (5a). C1 is characterized by the highest eBC (Fig. 5b), sulfate (total sulfate, particularly anthropogenic sulfate, Fig. 4h and Fig. 7d), ammonium (Fig. 4f) among the five clusters. These chemical species are usually abundant in *Arctic haze* (Law and Stohl, 2007; Quinn et al., 2007) and found to be mainly related to the accumulation mode aerosol (Tunved et al., 2013; Porter and

Clarke, 1997; Radke et al., 1984). This is supported by the highest concentrations in the accumulation mode of C1 compared to the other clusters (Fig. 2b). Thus, C1 is identified as anthropogenic *Arctic haze*.

#### 4.1.2 C2: Anthropogenic wintertime *Arctic haze*

C2 shows a similar size distribution to that of C1 (Figs. 2a) but with a much lower concentration in the accumulation mode, which is still higher than those in the other clusters (Figs. 2b). Compared to the other clusters, C2 has intermediate

concentrations of eBC (Fig. 5b), sulfate (particularly anthropogenic sulfate, Fig. 7d) and ammonium (Fig. 4f). The main differences relative to C1 are: a temporal trend towards colder January-March months (Fig. 2c) and abundance of sea salt. PSCF maps also point to the Greenland coast and Northern Europe as potential source regions (Fig. 8), likely associated with industrial activities. In summary, whilst C1 is mainly associated with the typical Arctic haze with sulfate and eBC-rich aerosols





(9.3%), industrial activities may also be an important source of anthropogenic aerosols (17.6%) in winter in the Arctic. In
addition, wintertime Arctic haze at the site is more likely mixed with SSA since the occurrence fraction of C2 is above twice
that of C1.

### 4.1.3 C3: Sea spray aerosol

The average size distribution of C3 is similar to that of sea salt species (e.g., $Na^+$, $Cl^-$ and $Mg^{2+}$), which are generally
dominated by a supermicron mode (Morawska et al., 1999; Porter and Clarke, 1997) with a peak at ∼4 $\mu$m in summer
(Dall'Osto et al., 2006; Ricard et al., 2002; Wall et al., 1988) and a peak between 2 and 3 $\mu$m (more close to 3 $\mu$m) in the
winter in the marine environment (Ricard et al., 2002). In addition, the average size distribution of C3 (Fig. 2a-b) agrees well
with that of sea salt aerosol observed in Antarctic (Fan et al., 2021). High loadings of sea salt (Fig. 7e) and their associated
species (Fig. 4c-e) and the average ratio of $Na^+$/$Ca^{2+}$ (Fig. 6) for C3 further indicate that C3 is associated with SSA. MSA
and apportioned bio-$SO_4^{2-}$ are also enriched in C3 (Fig. 4k and Fig. 7c).
Arctic MSA has been reported to be abundant in late spring and in summer (Becagli et al., 2019; Sharma et al., 2019), but
the C3 does not show the same seasonality. This inconsistency might be due to the mixture of sea spray and biogenic sources.
Ricard et al. (2002) reported that MSA has two submicron modes (∼0.3 and ∼0.7 $\mu$m) and one supermicron mode (∼2 $\mu$m)
between 43 nm and 20 $\mu$m in summer of the marine environment. These modes are in agreement with those for wintertime
sea salt aerosol (Ricard et al., 2002). The PSCF map also shows that C3 aerosols are likely coming from surrounding ocean
and sea ice (Fig. 8c), where sea salt and MSA are originated. In addition, the probability density distribution of temperature
for C3 presents a bimodal pattern (Fig. A4) with two modes of cold temperature at ∼-10 °C and warm temperature at ∼5 °C,
respectively. Thus, this cluster is associated with sea spray aerosols coming from marine open ocean, likely co-existing with
secondary MSA-containing biogenic aerosol.

### 4.1.4 C4: Unidentified sea spray-related aerosol

The average size distribution of C4 is quite similar to that of C3, but with an enhancement at larger diameters and a lower
average ratio of $Na^+$/$Ca^{2+}$ (Fig. 6). The main differences relative to C3 are: more sea salt mass (Fig. 2a and Fig. 7e), more
source contribution from Greenland (Fig. 8d) and less source contribution from the open ocean (Fig. 8d). The PSCF map for
C4 is quite different compared to C3 (Fig. 8), suggesting that drivers of C4 might be different from those of C3. Specifically,
potential sources of C4 are in shorter distance than C3 and are observed largely from Greenland (Fig. 8), which is mostly
covered by snow throughout the year.
The ratio of $Ca^{2+}$/$Na^+$ suggests that C4 is more likely associated with SSA compared to mineral dust (Fig. 6). Note that a
few samples of C4 in Fig. 6 tend to be mineral dust, especially for samples with high concentration of $Ca^{2+}$. This may suggest
a possible source of uplifted snow with mineral dust that was deposited on the snow surface. C4 also shows a higher relative
abundance in the months of September-November (Fig. 2c). Therefore, C4 is considered as unidentified sea spray aerosol,
possibly short-distance sea spray or a mixture of sea spray and blowing snow. More research is needed to better understand the
potential sources of C4.





### 4.1.5   C5: Mineral dust

C5 has distinct peaks in the coarse modes (Fig. 2a-b) pointing towards dust sources (Denjean et al., 2016; Ricard et al., 2002; Porter and Clarke, 1997). C5 occurs usually from June to October, which are the months with frequent dust emissions from Svalbard and/or high-latitude sources (Tobo et al., 2019). The ratio of $Ca^{2+}/Na^+$ (Fig. 6) and the mineral fraction of sulfate (Fig. 7b) for C5 further indicate a mineral dust source. According to the PSCF map (Fig. 8e), C5 is very likely originated from typical dust sources at high-latitude ice free terrain (proglacial fields or floodplains where fine glaciofluvial sediment deposits are exposed to wind), such as western/northern coast of Greenland (e.g., Kangerlussuaq area), northern Eurasia and coastal Alaska (Tobo et al., 2019; Groot Zwaaftink et al., 2016; Bullard et al., 2016; Crusius et al., 2011). Thus, C5 is identified as mineral dust.

### 4.2   Drivers of coarse-mode aerosols in the Arctic

In the present study, Arctic aerosol is characterized by prevalent *Arctic haze* between winter and late spring, biogenic aerosol and mineral dust during warm seasons and sea spray aerosol throughout all seasons. The seasonal cycle in the Arctic aerosol is driven by many aspects, such as the annual cycle in Arctic sea ice, temperature, radiation, atmospheric oxidants, cloud properties, seasonally varying transport and removal mechanisms (Abbatt et al., 2019; Willis et al., 2018). In this study, potential drivers of the coarse-mode anthropogenic aerosol and natural aerosol are discussed in sections 4.2.1 and 4.2.2.

### 4.2.1   Coarse-mode anthropogenic aerosol

High occurrences of C1-2 in winter and early spring suggest that anthropogenic sources are a predominant contributor to *Arctic haze*, particularly for accumulation-mode aerosol (Fig. 2a-b). According to the chemical composition of C1-2, *Arctic haze* contains a large fraction of eBC (Fig. 5b), sulfate (particularly anthropogenic sulfate, Fig. 4h and Fig. 7d), ammonium (Fig. 4f), oxalate (Fig. 4j), implying that both primary and secondary sources (including secondary inorganic and organic aerosols) contributed to the *Arctic haze*. On average, the contributions of anthropogenic sources to total sulfate were 91.7% for C1 and 84.1% for C2, which are almost twice as high as for the other clusters (ranging from 34.2% for C5 to 52.7% for C3) (Fig. 7). According to the PSCF maps of C1-2 (Fig. 8a-b), long-range transport from northern Eurasia to Arctic regions might be mainly responsible for the anthropogenic aerosol in winter and spring. It is suggested that meteorological conditions during the *Arctic haze* period are conducive to long-range transport from northern Eurasia to the Arctic (Willis et al., 2018; Law and Stohl, 2007). Furthermore, the polar front could extend to about 40°N in winter to include industrial emissions that can be transported into the high Arctic, thus leading to the presence of industrial sources in the wintertime Arctic aerosol (Law and Stohl, 2007).

However, C2 contains a sea salt fraction (Fig. 6-7). Prather et al. (2013) showed that submicron sea spray aerosol consists of two externally mixed particle types, sea salt mixed with organic carbon and organic aerosol without chloride, based on transmission electron microscopy with energy-dispersive X-ray analysis. Yu et al. (2019) found that sulfate and organic matter are often internally mixed based on individual particles (100 nm to 2 μm) collected in the Svalbard Archipelago in summer.





Thus, the enrichment of eBC, sulfate, organic aerosol and sea salt in C2 is likely due to the chemical mixing state of the particle
size ranges, leading to a mixed anthropogenic and natural aerosol, which is a transition mode driven by both *Arctic haze* and
sea spray.

### 4.2.2 Coarse-mode natural aerosol

The drivers of natural aerosol in the Arctic are even more complex than that of anthropogenic aerosol mainly because of various
natural sources. Our analysis identified three main natural sources, including open ocean sea spray aerosol (Figs. 2a and 6 and
7e and 8c), unidentified source of sea pray-related aerosol (Figs. 2a and 6 and 7e and 8d) and mineral dust (Figs. 2a and 6 and
7 and 8e). These natural aerosols dominated the Arctic aerosol population with a fraction of >50% throughout all seasons, and
with a much higher fractions of >80% in summer and autumn than those during the *Arctic haze* period (Fig. 2b). Warm seasons
facilitate the production of natural aerosols in many ways, such as enhanced marine primary production (Becagli et al., 2016),
biogenic new particle formation and growth (Lange et al., 2019, 2018; Dall'Osto et al., 2018, 2017) as well as high-latitude
dust source emissions (Tobo et al., 2019; Groot Zwaaftink et al., 2016; Bullard et al., 2016).

Notably, C3-4 comprised the largest fraction of the aerosol population throughout all seasons (Fig. 2b), suggesting that SSA
is ubiquitous in the high Arctic regardless of season. The production of sea salt has been attributed to various mechanical
processes, such as wind-driven sea spray and bubble bursting from open ocean (Quinn et al., 2015) or leads (Kirpes et al.,
2019; May et al., 2016) or wind-blown snow (Huang and Jaeglé, 2017). The prevalence of C3 in all seasons and the closest
ratio of $Ca^{2+}$/$Na^+$ to sea water (Fig. 6) provide a compelling evidence that it is driven by open ocean emission. More studies
are needed to elucidate the underlying drivers of the occurrence of C3 throughout all seasons.

MSA-containing biogenic aerosol was observed mainly in C3-4 (Fig. 4k and Fig. 7c). Lin et al. (2012) found that MSA
constitutes a major fraction of total aerosol sulfate over the remote ocean by means of isotope analysis. However, their analysis
was based on submicron-mode (<1 $\mu$m) aerosol. Our study suggests that a fraction of MSA could occur in the coarse mode and
is not entirely in the submicron fraction. The fractions of biogenic sulfate in total sulfate for C3 (~24.6%) and C4 (~21.2%)
were much higher than those for anthropogenic *Arctic haze* (i.e., ~3.3% for C1 and ~3.4% for C2), but were lower than that
of anthropogenic sulfate (i.e., 53.7% for C3 and 50.3% for C4). The contributions of biogenic emissions to sulfate for C3-4 are
consistent with Udisti et al. (2016) who reported that biogenic emissions contributed ~35% to sulfate in summertime Arctic,
mainly because C3-4 predominated (>80%) summertime Arctic aerosol. However, the biogenic sulfate fraction for C3 is less
than half of that from anthropogenic sulfate. It means that long-range transported anthropogenic pollution still appears to be
the largest source of sulfate in the Arctic, even for MSA-containing biogenic aerosol type.

Biogenic aerosol in the Arctic is likely to be mixed with SSA, which is consistent with Prather et al. (2013) which reported
that supermicron marine biogenic aerosol is mixed with SSA based on the single particle chemical mixing state.

The seasonal variation in biogenic aerosol in the Arctic appears to be related to sea ice retreat, but the interannual trend in
MSA concentration is likely to be driven by the extent of the ice-free marginal zone (Becagli et al., 2019, 2016).

Mineral dust was identified as another natural aerosol mainly in C5. Mineral dust aerosol appears to be originated from
dust sources at high-latitude ice free terrain (Fig. 8e) in warm seasons (Fig. A4d). Unfortunately, geological minerals were not



reconstructed in the present study due to lack of measurements of mineral elements. Measurements of metal elements at the station are urgently needed in future studies to better understand the mineral dust in the high Arctic (Becagli et al., 2020).

## 5 Conclusions


In summary, we have reported relatively long-term (2015-2019) measurements of aerosol aerodynamic volume size distributions up to $20\mu$m in the high Arctic for the first time. Our results provide insights into supermicron aerosol properties and their potential sources around at an Arctic site in Svalbard, particularly sea spray aerosol, MSA-containing biogenic aerosol and mineral dust. The study elucidated five main aerosol volume size distributions and their natural and anthropogenic sources in

the high Arctic, which may help to better understand the complex interactions and feedbacks between aerosol, cloud, radiation and air-sea dynamic exchange and biota (Abbatt et al., 2019; Willis et al., 2018; Browse et al., 2014). Our study shows that about two third of the coarse-mode aerosols are related to two sea spray-related aerosol clusters, indicating that sea spray aerosol - commonly assumed to be a known source from open ocean - may be more complex in the Arctic environment. Further studies - both ambient and laboratory based - are strongly needed to understand sea spray sources. In addition, measurements of metal

elements up to coarse mode are encouraged in future studies to better understand local and transported mineral dust as well as anthoropogenic pollution (e.g., industrial activity and shipping emission) in the high Arctic.

*Data availability.*  The APS data can be accessed from Traversi et al. (2020). The absorption coefficient data are available upon request from Gilardoni et al. (2020). Data supporting this publication can be accessed upon request from the corresponding authors.

*Author contributions.*  MDO conceived the study and discussed the analysis and results. ZS supervised the project and funded this study.

CS performed data analysis, data visualization, data interpretation and wrote the original draft. DCSB provided the raw R codes of $k$-means clustering. AL, MM, RT, SB and SG carried out the field measurements at the GVB station and chemical analysis of the offline samples and provided feedbacks on the draft. JS, KEY, MDO, ZS, JB and RMH participated in editing the manuscript. All the authors commented on the manuscript and approved the submission.

*Competing interests.*  The authors declare that they have no competing interests.

*Acknowledgements.*  This work is funded by UK Natural Environment Research Council (NE/S00579X/1). The authors acknowledge the staffs of the Arctic Station Dirigibile Italia of the National Research Council of Italy for their support in measurements at the GVB station. JS holds the Ingvar Kamprad Chair for Extreme Environment Research. The authors acknowledge the NOAA Air Resources Laboratory (ARL) for providing the HYSPLIT model to analyze the back trajectories.



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

**Figure 1.** Monthly average aerosol volume size distribution from 2015 to 2019. Missing data in some months but good data coverage from March to June. No data available in December. Monthly average aerosol number size distribution measured by the APS during 2015-2019 is shown in Fig. A1.

**Figure 2.** (a) Normalized and (b) absolute aerosol volume size distributions for the five clusters. Shaded area for the size distribution shows one standard deviation. (c) Seasonal variation in relative abundance of the five clusters. Interannual variation in the relative abundance from 2015 to 2019 in the months of (d) March, (e) April, (f) May and (g) June. We only show the interannual variation from March to June because these months have good data coverage in 2015-2019.

**Figure 3.** (a) Diurnal variation of the mean of the aerosol volume size distribution and (b) polar plots of total volume concentration for the five aerosol clusters. Polar plots: average concentrations are shown to vary by wind speed and wind direction. The color scale of the polar plot is the average concentration level normalized by dividing by the mean value from each cluster.

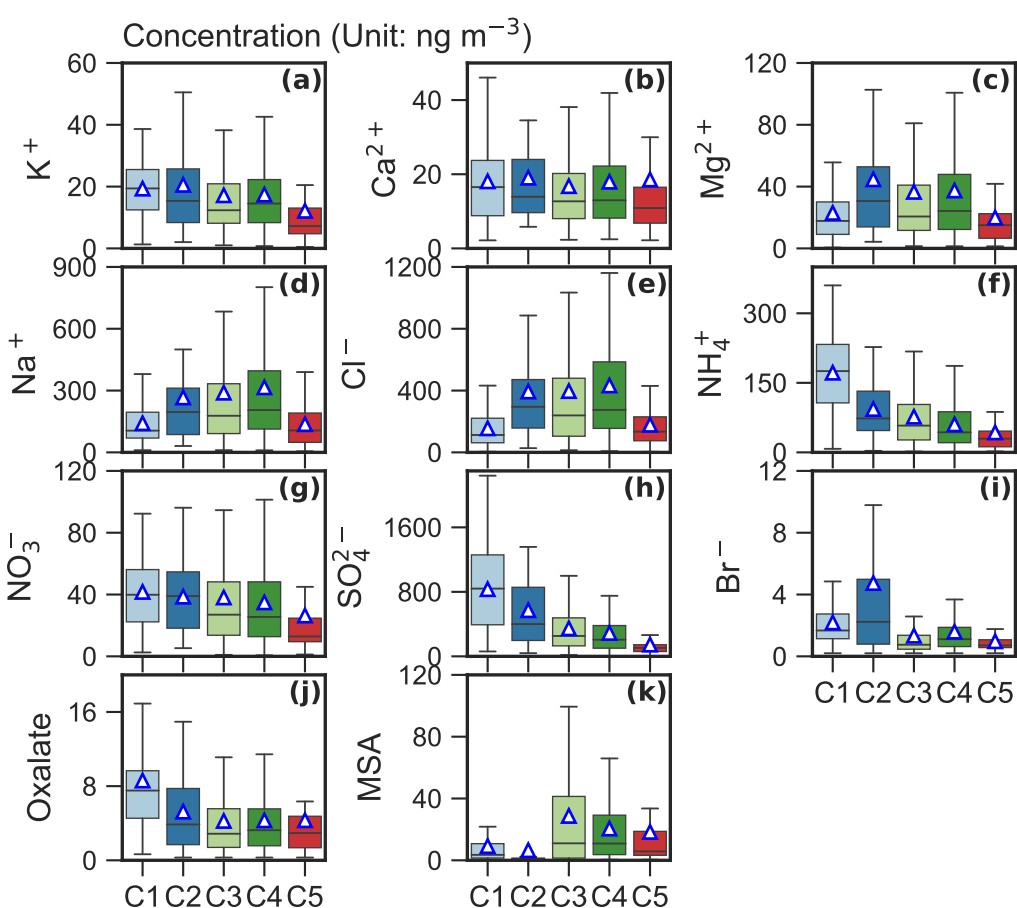

**Figure 4.** Box plot of mass concentration (unit: ng m$^{-3}$) of each chemical species for the identified clusters. The mean values are marked as triangles.





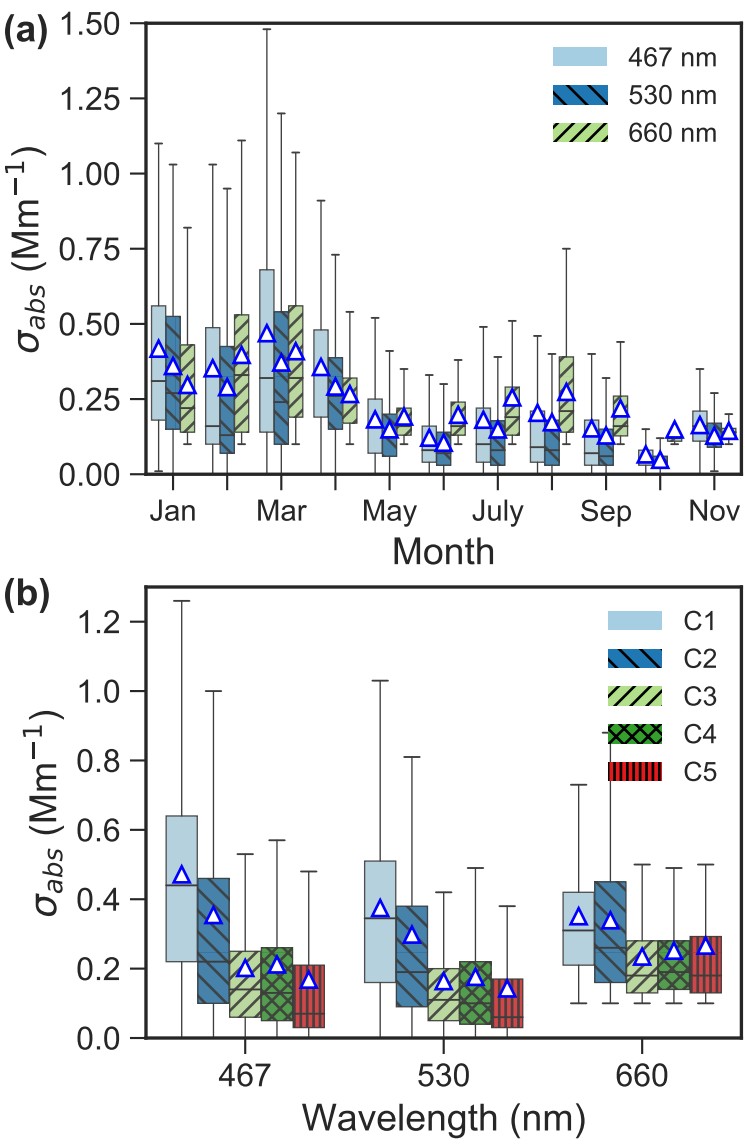

**Figure 5.** (a) Seasonal variation of absorption coefficient ($\sigma_{abs}$) at wavelengths of 467 nm, 530 nm and 660 nm in 2015-2019 and (b) boxplot of absorption coefficient at the three wavelengths for each cluster. The mean values are marked as triangles.



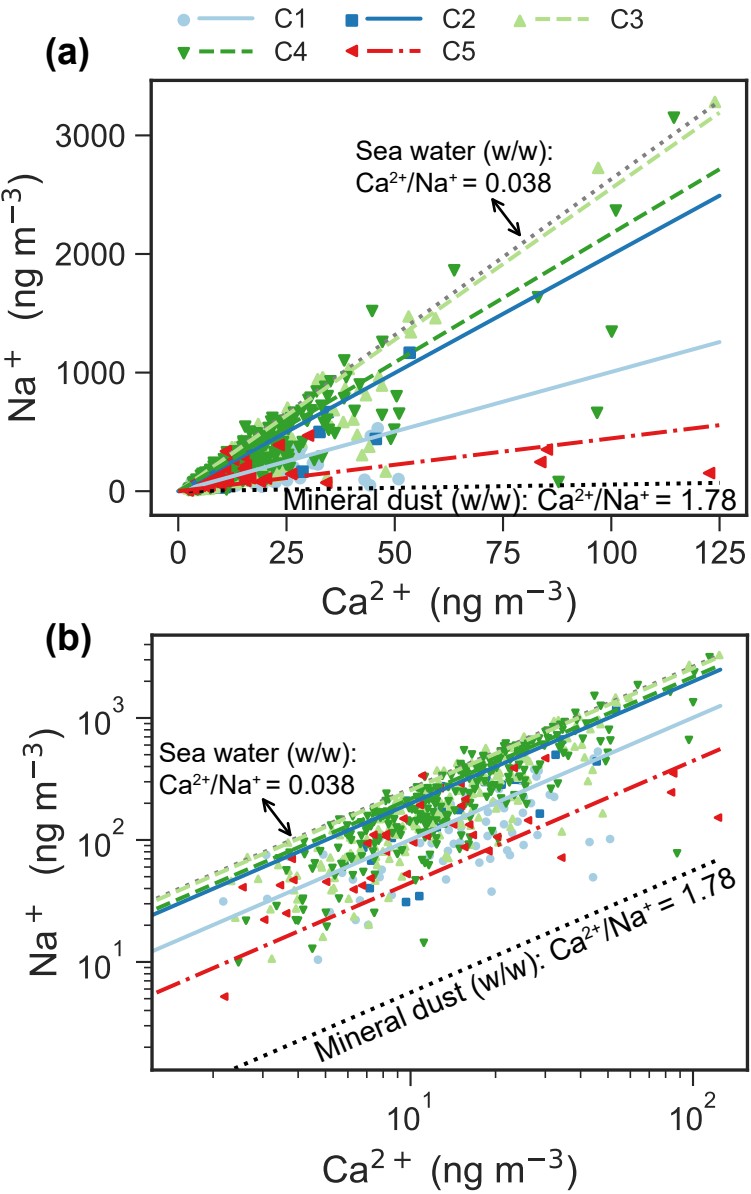

**Figure 6.** Scatter plot (a: linear scale, b: logscale) of daily $Na^+$ versus $Ca^{2+}$ mass concentrations for each cluster. The ratios of $Ca^{2+}/Na^+$ from sea water and Earth's crust are denoted by dotted lines. The scatter points are usually in the ranges of ratios of sea water and Earth's crustal (Bowen et al., 1979). The reduced major axis (RMA) regression between $Na^+$ and $Ca^{2+}$ for each cluster is superimposed on the figure. The parameters of the regressions are listed in Table A1.





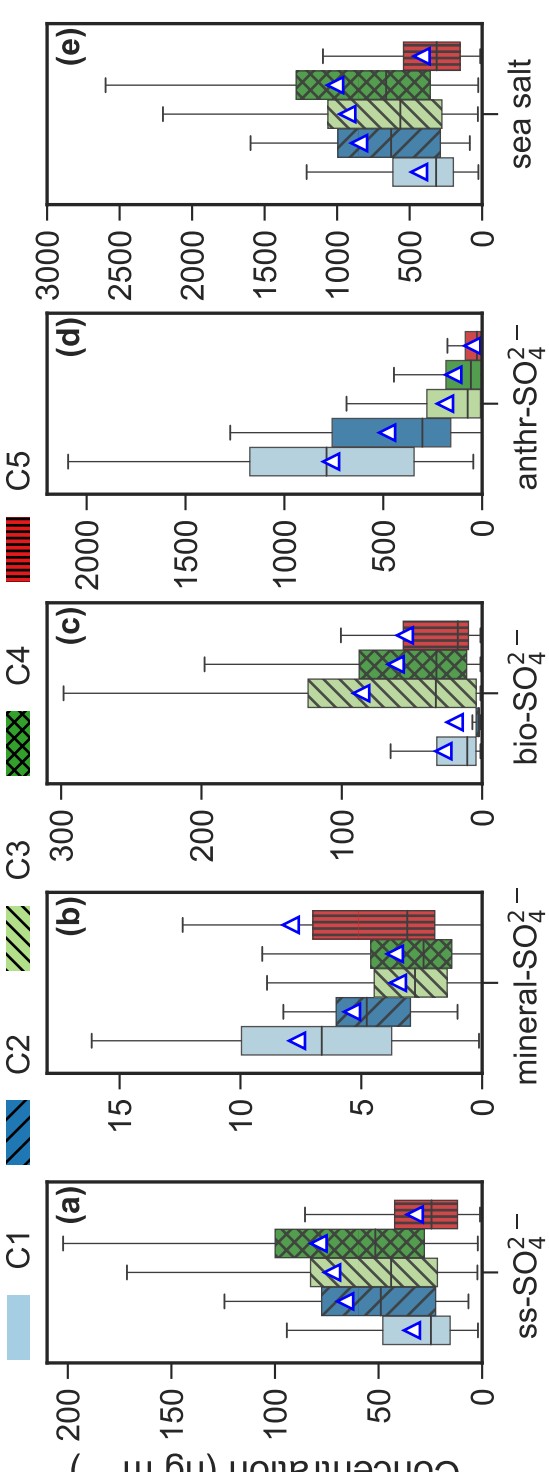

**Figure 7.** Box plot of (a) sea-salt $SO_4^{2-}$ (ss-$SO_4^{2-}$), (b) mineral $SO_4^{2-}$ (mineral-$SO_4^{2-}$), (c) biogenic $SO_4^{2-}$ (bio-$SO_4^{2-}$), (d) anthropogenic $SO_4^{2-}$ (anthr-$SO_4^{2-}$) and (e) sea salt for each cluster. The (e) sea salt aerosol is calculated by multiplying the ss-$Na^+$ by a factor of 3.248 (Kerminen et al., 2000; Brewer, 1975). The mean values are marked as triangles.



**Figure 8.** Potential source contribution function (PSCF) map for each cluster and all data. The GVB station is denoted as a black dot on the map.





**Table 1.** Occurrence and key species for the five clusters.

|     | Occurrence   | Key species                                    |
| --- | ------------ | ---------------------------------------------- |
| C1  | 9.3±5.1%     | eBC; Sulfate; Ammonium; Oxalate                |
| C2  | 17.6±17.4%   | eBC; Sulfate; Ammonium; Sea salt species*      |
| C3  | 34.2±8.2%    | Sea salt species*; MSA;                        |
| C4  | 32.0±18.7%   | Sea salt species*; MSA                         |
| C5  | 6.9±5.5%     | High ratio of $Ca^{2+}/Na^{+}$                 |

Note. *Sea salt species here refer to $Na^{+}$, $Cl^{-}$ and $Mg^{2+}$.



**Figure A1.** Monthly average aerosol number size distribution from 2015 to 2019. Missing data in some months but good data coverage from March to June. No data available in December.



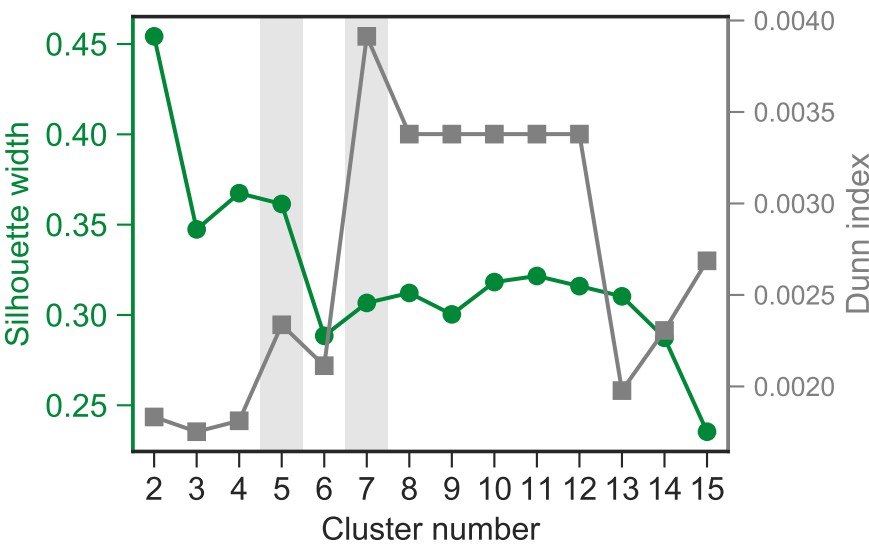

**Figure A2.** Silhouette width and Dunn index for cluster numbers ranging from 2 to 15.

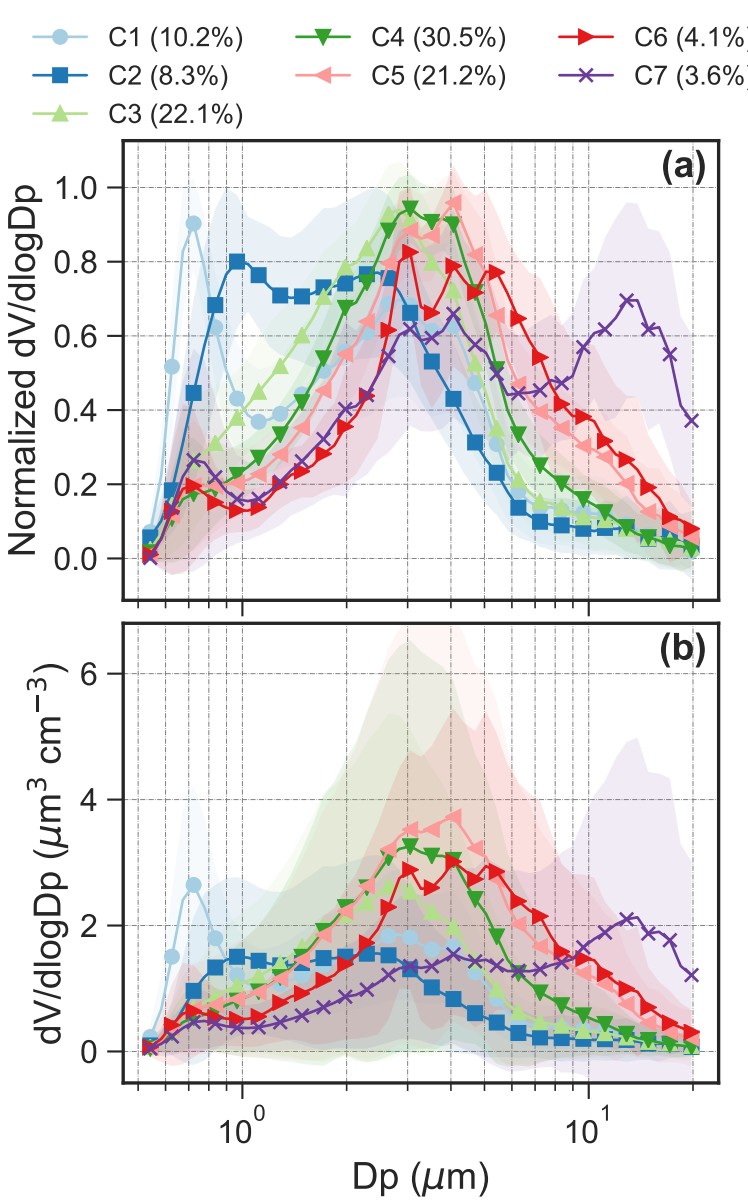

**Figure A3.** Average aerosol size distribution of the seven clusters using a seven-cluster solution.

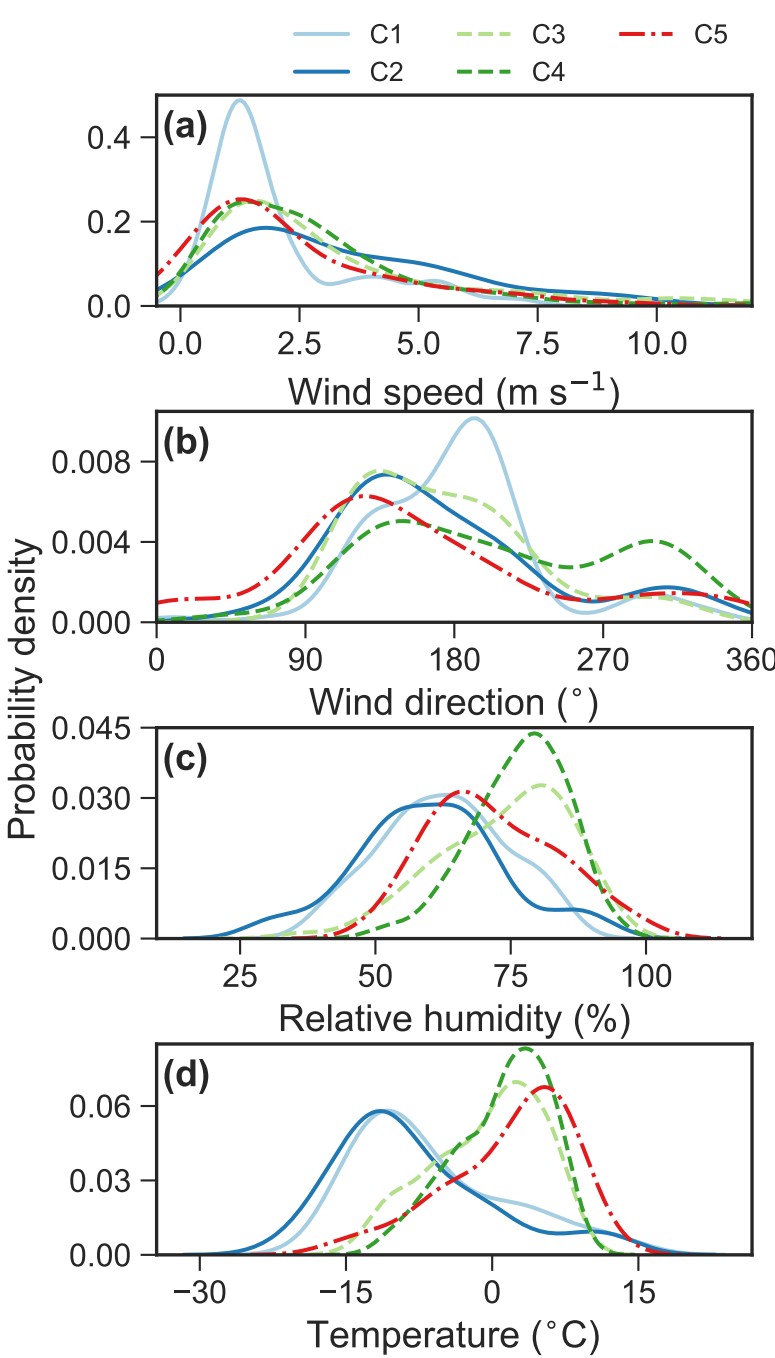

**Figure A4.** Probability density distribution of the meteorological parameters for each identified cluster.





**Table A1.** Parameters for reduced major axis (RMA) regression between $Na^+$ and $Ca^{2+}$ for the five clusters

|  | C1 | C2 | C3 | C4 | C5 |
|---|---|---|---|---|---|
| n | 68 | 22 | 185 | 256 | 39 |
| r-square | 0.33 | 0.78 | 0.84 | 0.7 | 0.16 |
| Slope | 10.1 | 19.9 | 25.5 | 21.7 | 4.5 |
| 2.5%-Slope | 8.2 | 16.0 | 24.1 | 20.3 | 3.3 |
| 97.5%-Slope | 12.3 | 24.8 | 27.1 | 23.2 | 6.0 |
| Intercept | -40.5 | -113.4 | -139 | -75.0 | 54.6 |
| 2.5%-Intercept | -80.8 | -206.0 | -164.8 | -102.3 | 26.0 |
| 97.5%-Intercept | -7.6 | -39.0 | -114.6 | -49.5 | 75.7 |