# Peer review of "Differentiation of coarse-mode anthropogenic, marine and dust particles in the high Arctic Islands of Svalbard"

_Atmospheric Chemistry and Physics, 2021_

## Referee Comment (RC1)

Comments:

This manuscript describes a cluster analysis of approximately five years (from 2015 to 2019) aerodynamic volume size distribution with diameter ranging from 0.5 to 20 μm at the Gruvebadet Observatory in the High Arctic Islands of Svalbard. Furthermore, the aerosol size distributions are complemented by aerosol chemical composition data. This work clearly distinguished the Arctic coarse-mode aerosols originated from anthropogenic sources (related to Arctic Haze) and from natural sources (related to open ocean and mineral dust). This result is an important and really interesting conclusions that has not been made in the past, because most of the previous studies have been only focused on investigating number size distributions of particles smaller than 1 μm, which measured by SMPS system. Overall, this manuscript is well written, and presents the main conclusion obviously. I support publication in ACP once the following issues can be addressed.

**Comments:**

Page 10 and Line 261: The present study suggests that the size distribution of C3 is associated with sea aerosol coming from the ocean. Typically, sea spray aerosol is generated *via* bubble-bursting processes under sufficient wind speed conditions. At wind speeds greater than approximately 5 m s$^{-1}$, breaking waves are formed on ocean surface. In this study, Fig. A4 shows density distribution of the wind speeds for the C3. However, it seems that wind speeds for C3 is not enough to produce the sea spray aerosol.

---

## Author Comment (AC1)

MS No. acp-2021-94

Title: Differentiation of coarse-mode anthropogenic, marine and dust particles in the high Arctic Islands of Svalbard

Authors: Congbo Song et al.

**Response to reviewers**

Reviewer #1

General comments: This manuscript describes a cluster analysis of approximately five years (from 2015 to 2019) aerodynamic volume size distribution with diameter ranging from 0.5 to 20 μm at the Gruvebadet Observatory in the High Arctic Islands of Svalbard. Furthermore, the aerosol size distributions are complemented by aerosol chemical composition data. This work clearly distinguished the Arctic coarse-mode aerosols originated from anthropogenic sources (related to Arctic Haze) and from natural sources (related to open ocean and mineral dust). This result is an important and really interesting conclusions that has not been made in the past, because most of the previous studies have been only focused on investigating number size distributions of particles smaller than 1 μm, which measured by SMPS system. Overall, this manuscript is well written, and presents the main conclusion obviously. I support publication in ACP once the following issues can be addressed.

Minor comments: The present study suggests that the size distribution of C3 is associated with sea aerosol coming from the ocean. Typically, sea spray aerosol is generated via bubble-bursting processes under sufficient wind speed conditions. At wind speeds greater than approximately 5 m s-1, breaking waves are formed on ocean surface. In this study, Fig. A4 shows density distribution of the wind speeds for the C3. However, it seems that wind speeds for C3 is not enough to produce the sea spray aerosol.

**Response:** We agree that sea spray aerosol is typically generated at high wind speed. In Fig. A4, we presented density distribution of wind speeds for the C3. We should have made it clear that the meteorological data are from a local meteorological station, which only represent local conditions but not long-range transport. Interestingly, the concentrations of C3 are generally higher at high wind speeds than that at low wind speeds (Fig. 3). To better illustrate the impacts of meteorological conditions on different aerosol clusters, we looked at the density distribution of the meteorological conditions for volume concentrations exceeded their 90-percentile values within each cluster and updated the Fig. A4 (see the below figure). The new figure shows that the average wind speeds for C3 exceeded their 90-percentile value have bimodal distribution with peaks around 2m/s and 10m/s, which may suggest that local and long-range transported sea spray aerosol are both important. We update relevant contents regarding Fig. A4 and added a sentence in Line 295-296.

[Figure]

Fig. R1 Probability density distribution of the meteorological conditions for aerosol volume concentrations exceeded their 90-percentile values within each cluster.

Reviewer #2

General comments:

This study focuses on coarse-mode aerosol particles and discusses their possible sources based on cluster analysis, aerosol chemical composition analysis, and air-mass backward trajectory analysis. The authors categorize the coarse particles into 5 clusters based on the difference of the spectra of their number-size distribution, and then suggest that the clusters C1 and C2 (27%) and the clusters C3, C4, C5 (73%) are attributable to anthropogenic and natural origins, respectively. Because previous studies have overlooked the possible importance of coarse-mode aerosols in the Arctic, the datasets presented here would be valuable for the science community. However, the methods and discussion on the source of coarse-mode aerosols might be biased, because there are some major problems with the methodology. As a result, it is hard for me to believe the percentages presented here (e.g., Lines 5-10) are indeed valid. I would like to ask the authors to answer the following comments and clearly explain the possible biases of their approaches.

**Response:** We have addressed all the major concerns below. Please see point by point response.

**Major Comments**

Based on the cluster analysis of the number-size distributions of coarse-mode aerosols (0.5 to 20 um), the authors divided them into 5 categories. I can agree that this categorization is one of the reasonable approaches for understanding the characteristics of spectra of their number-size distributions. The authors also explained that the percentage of the clusters C1, C2, C3, C4, and C5 are 9%, 18%, 34%, 32%, and 7%, respectively (Lines 8-10; Table 1). However, it seems that there is a large seasonal bias of the available data (Figure 1). In particular, the wintertime data are very lacking. The authors would need to explain the possible influence of this bias on their cluster analysis.

**Response:** We agree with the reviewer 2 and it is indeed important to stress we only quantify sources of particles for a specific period of the year - from March to October. We do not have data from the winter. We have made this point very clear in our manuscript. We added a sentence in Line 6-7 to stress that the percentages presented in our study are "during the study period (mainly from March to October)".

We would like to emphasize that we did report the full monthly data and the average±standard deviation for the study period (as shown in the below Table R1). The uncertainty due to seasonal variation were estimated by the standard deviation in Line 200-218 and Table 1. We did not report the standard deviation in the abstract.

Table R1: Monthly occurrence frequency of the five clusters

|  | C1 | C2 | C3 | C4 | C5 |
|---|---|---|---|---|---|
| **Jan** | 8.7% | 43.2% | 47.1% | 1.0% | 0.0% |
| **Feb** | 6.8% | 48.8% | 30.2% | 9.0% | 5.2% |
| **Mar** | 15.9% | 32.2% | 34.7% | 15.6% | 1.6% |
| **Apr** | 18.8% | 16.7% | 39.4% | 24.0% | 1.0% |

| | | | | | |
|---|---|---|---|---|---|
| **May** | 8.5% | 9.3% | 42.5% | 37.2% | 2.6% |
| **June** | 4.4% | 4.0% | 38.3% | 44.5% | 8.9% |
| **July** | 6.5% | 8.4% | 32.6% | 42.0% | 10.5% |
| **Aug** | 10.3% | 2.3% | 25.3% | 51.4% | 10.8% |
| **Sep** | 10.8% | 4.5% | 33.1% | 37.7% | 13.8% |
| **Oct** | 1.9% | 6.6% | 19.2% | 57.3% | 15.0% |
| **Mean** | 9.3% | 17.6% | 34.2% | 32.0% | 6.9% |
| **Standard deviation** | 5.1% | 17.4% | 8.2% | 18.7% | 5.5% |

Although the authors describe that they have used the aerosol chemical composition data from several online and offline measurements (Lines 5-6), these datasets are mainly based on ion chromatography (i.e., water-soluble particles) and PSAP (i.e., black carbon). For this reason, the results presented here would underestimate the contribution of "insoluble" particles to the population of coarse-mode aerosols. For example, although they discuss the contribution of mineral dust particles based on Ca content (Ca/Na ratio), Ca-containing components (e.g., calcite, dolomite) are not necessarily major mineral components. Actually, the mineral composition analysis of mineral dust particles in high latitudes using SEM/EDS and/or XRD analysis (e.g., Moroni et al, 2015; Tobo et al., 2019; Sanchez-Marroquin et al., 2020; Adachi et al., 2021) has shown that mineral dust particles contained few or no Ca components. Also, the presence of insoluble organic particles and biological particles (e.g., bacteria, fungal spores, pollens) has been reported by filed measurements at the Gruvebadet and Zeppelin Observatories in Svalbard (Geng et al., 2010; Tobo et al., 2019). For these reasons, additional analysis of insoluble particles would be required if they would like to quantify the dominant aerosol types in each cluster (C1 to C5).

**Response:** Yes, ideally, we have data on "insoluble" fractions which will further support the data interpretation for this cluster. The main objective of the manuscript was to illustrate the major categories of coarse-mode particle size distribution. The chemical data were thus used to support identifying possible sources. If we are estimating the absolute dust concentration, we will have to measure other components of crustal elements. But in this case, the presence of $Ca^{2+}$ suggest dust contribution to the cluster 3. Since we were comparing the non-sea salt $Ca^{2+}$ contents between clusters, thus the absolute concentration of mineral dust in each cluster is less important (and not quantified). There is a good correlation between insoluble microparticles and $Ca^{2+}$ concentrations in ice cores in Greenland and some authors suggested that the $Ca^{2+}$ concentration is a reasonable proxy for the insoluble-mineral-dust variability (Ruth et al., 2002, Kang et al., 2015). We do recognize not all Ca in high latitude dust may be soluble (and thus measurable as $Ca^{2+}$) even though there are significant amount Ca in some of the high latitude dust, such as from the Iceland (Baldo et al., 2020). Therefore, $nss-Ca^{2+}$ is a conservative tracer for dust. We added a sentence in Line 143-146: It is worth noting that Ca in high latitude dust may not be in the form of carbonate so not measurable as $Ca^{2+}$ (Baldo et al., 2020; Bachelder J et al., 2020; McCutcheon, J et al., 2021). Some studies also reported few Ca-containing high latitude dust (Baldo et al., 2020; Bachelder J et al., 2020; McCutcheon, J et al., 2021). Thus, the $nss-Ca^{2+}$ is a conservative tracer for mineral dust. We were not able to distinguish biogenic fractions in the clusters (e.g., bacteria, fungal spores,

pollens), which have been reported by field measurements in Svalbard (Geng et al., 2010).

I doubt if each category was indeed characterized by anthropogenic or natural sources. As far as I check the results in Figures 4 and 7, each cluster were likely to be influenced by both anthropogenic and natural origins in most cases. For example, the aerosol population in the clusters C1 and C2 might be largely influenced by Arctic haze (Lines 244-261); however, I cannot think that the influence of natural aerosols on the population of coarse-mode aerosols was negligibly small. Although the clusters C3 and C4 are categorized as sea spray aerosols (Lines 262-291), a case study at the same site (Gruvebadet Observatory) in summer (Geng et al., 2010) demonstrated that both sea salt and mineral dust particles were always detected in the same samples. It is also skeptical that the cluster C5 can be identified as mineral dust only (Lines 292-300).

**Response:** Cluster analysis is a semi-quantitative study of APS size distributions, where time trends were analyzed. Complementary information is used to further specify the cluster types. We agree that each cluster cannot be solely attributed to one source only, but this is expected in these types of semi-quantitative studies. Source apportionment studies with K-means have intrinsic uncertainties. Nevertheless, this has been shown to be relatively successful in the Arctic air, which are usually much simpler than in an urban atmosphere. We believe that our analysis shed new light on sources and shows interesting trends that are important and only achievable with the APS observations. Detailed responses are as follows:

1. The analysis gave average occurrence fractions of coarse-model aerosols dominated by anthropogenic and nature sources, which is basically unknown to date and can only achievable by k-means.

2. For daily particle size distribution, it is normal to see mixing sources in the identified clusters, which is similar to single particles collected by cascade impactors. However, we didn't intend to separate the mixing sources in current analysis, but it will be our future study.

3. We added a caveat for interpretating the sources of the identified clusters. Line 258-261: changed from "The potential sources of the aerosol clusters were identified based on (1) volume size distribution, (2) chemical components, (3) seasonality in occurrence of the clusters and (4) potential source maps." to "The dominant source for each cluster that distinguishing the aerosol clusters from each other was identified based on (1) volume size distribution, (2) chemical components, (3) seasonality in occurrence of the clusters and (4) potential source maps. It is possible that each cluster does not represent a pure source but contain signals of other aerosol sources. In the following contents, sources of aerosol clusters represent the sources that distinguishing the aerosol clusters from each other.".

4. The title of the section 4.1 was changed to "Dominant sources of the aerosol clusters".

5. We added a sentence in Line 384-385: Furthermore, mineral dust is often mixed with sea salt in C5, which makes it difficult to quantify the absolute contributions from dust to C5. And Line 398-399: More advanced receptor modelling and observational data, such as insoluble particles are needed to address the mixing sources found in the k-means clusters.

There would be some problems with the use of backward trajectory analysis of air masses. Although the authors discuss the possible source of BC, sea spray, and/or mineral dust based on the trajectory analysis, their emission in high latitudes would be largely altered by the seasonal variability of snow/ice cover over land and sea. The authors would need to check the variation of snow/ice cover as well as air mass history (Figure 8). The explanations presented in Lines 230-239 are insufficient. In addition, the trajectory analysis might overlook the possible contribution of local sources (namely, Svalbard and its surroundings).

**Response:** Figure 8 shows the potential source areas of the identified clusters. We agree that potential source regions of the clusters would vary the snow/ice cover over land and sea. We would like to point out that our Figure 8 shows the source regions based on multi-year and multi-season average data; it gives a general idea of where the aerosol came from. To illustrate the impacts of ground types on the identified clusters, we added additional analysis (see Figure R2) about Relative contribution of the accumulated time for back-trajectory air parcels spent over sea, sea ice, snow, land and above mixing layer (ML) for each month and each cluster with and without considering back trajectories above mixing layer in the revised manuscript. It should be noting that seasonal variation of the relative contributions from different surface types in each year of 2015-2019 are quite similar (see Figure R3).

[Figure]

Figure R2: (a) Relative contribution (%) of the accumulated time for back-trajectory air parcels spent over sea, sea ice, snow, land and above mixing layer (ML) for each month and each cluster. (b) Without considering back trajectories above mixing layer.

[Figure]

Figure R3: Relative contribution (%) of the accumulated time for back-trajectory air parcels spent over sea, sea ice, snow and land in each year of 2015-2019.

Line 202-204: The relative contribution from different regions to back-trajectory air parcels for C1 is similar to that in March and April, which is characterized by the highest fraction of historical air parcels above mixing layer height among the five clusters or all seasons (Fig. A4a).

Line 218: These months also received the highest fractions of historical air parcels spent above lands (Fig. A4a-b).

Line 319-321: The considerable fractions of air parcels spent over land and sea/sea ice for C5 (Fig. A4a-b) further indicate the possibility of mixture of mineral dust and sea salt, which has been reported by Geng et al. (2010).

Lines 230-239 are results from PSCF analysis, the explanations have been moved to the 'Discussion' section. It is possible that back trajectory analysis might overlook the possible contribution from local sources. Based on the current data and analysis, it is challenging to quantify the contributions from local sources, which is a subject for future research. We added a sentence that "Note that potentially local sources cannot be captured by the back-trajectory analysis." In Line 255.

Specific comments
1.   Line 30: Where is the Zeppelin Observatory?
**Response:** We added 'in Svalbard' in Line 31.

2.   Figure 3: Was there no difference in the diurnal variation of aerosol volume size distribution

between the periods of polar day (summer) and night (winter)?

**Response:** Please find the below Figure R4. It shows diurnal variation of the mean of the aerosol volume size distribution in summer (June-Sep) and winter (Nov-Feb) for the five clusters. Though there are differences in the diurnal variation of aerosol volume size distribution in summer and winter, the peak diameters of the diurnal variation in the two seasons are similar. Since we used daily average aerosol volume size distribution and daily chemical data, the differences between diurnal aerosol volume size distributions in summer and winter were not further discussed in our manuscript.

[Figure]

Figure R4: Diurnal variation of the mean of the aerosol volume size distribution in summer (June-Sep) and winter (Nov-Feb) for the five clusters

Lines 206-208: It was hard for me to understand this explanation. Why did you think that "the polar plots show that high concentrations of the five clusters were typically found when wind speed was higher than 5 m s−1 (Fig. 3a), indicating limited impact of local emissions on coarsemode particles at the GVB station except for wind lifted particles"?

**Response:** We changed this sentence to "The polar plots show that high concentrations of the five clusters were typically found when wind speed was higher than 5 m s-1 (Fig. 3a), which is consistent with the probability density distribution of the wind speed for aerosol volume concentrations exceeded their 90-percentile values within each cluster (Fig. A5)." in Line 221-224.

Sections 4.1.1 to 4.1.5: The title of these sections would not be appropriate. I would recommend the titles like "4.1.1 Possible sources of C1" or something like this. In addition, more detailed discussion on possible sources of each cluster would also be required in each section.

**Response:** The title of section 4.1 is "Identification of potential sources of the aerosol clusters", thus it would be unnecessary to use "Possible sources of" again for titles of subsections 4.1.1 to 4.1.5. We changed the title of section 4.1 from "Possible sources" to "Dominant sources". Hope that would be a more appropriate title for section 4.1.

Lines 281-282: Why do you think that the main difference of C4 relative to C3 is "less source contribution from the open ocean (Fig. 8d)"? It seems that most air masses originate from the North Atlantic Ocean, which is rarely covered with sea ice during all seasons.

**Response:** Sorry for the mistake. We checked the relative contribution from surface types for the clusters, the air mass origins for C4 are similar to that for C3 (see Fig. A4). We deleted this sentence.

**References**

Adachi, K., Oshima, N., Ohata, S., Yoshida, A., Moteki, N., & Koike, M. (2021). Compositions and mixing states of aerosol particles by aircraft observations in the Arctic springtime, 2018. Atmospheric Chemistry and Physics, 21(5), 3607-3626.

Geng, H., Ryu, J., Jung, H. J., Chung, H., Ahn, K. H., & Ro, C. U. (2010). Single-particle characterization of summertime Arctic aerosols collected at Ny-Ålesund, Svalbard. Environmental science & technology, 44(7), 2348-2353.

Moroni, B., Becagli, S., Bolzacchini, E., Busetto, M., Cappelletti, D., Crocchianti, S., ... & Vitale, V. (2015). Vertical profiles and chemical properties of aerosol particles upon Ny-Ålesund (Svalbard islands). Advances in Meteorology, 2015.

Sanchez-Marroquin, A., Arnalds, O., Baustian-Dorsi, K. J., Dagsson-Waldhauserova, P., Harrison, A. D., Maters, E. C., ... & Murray, B. J. (2020). Iceland is an episodic source of atmospheric ice-nucleating particles relevant for mixed-phase clouds. Science advances, 6(26), eaba8137.

Tobo, Y., Adachi, K., DeMott, P. J., Hill, T. C., Hamilton, D. S., Mahowald, N. M., ... & Koike, M. (2019). Glacially sourced dust as a potentially significant source of ice nucleating particles. Nature Geoscience, 12(4), 253-258.

Ruth, U., Wagenbach, D., Bigler, M., Steffensen, J. P., Röthlisberger, R., & Miller, H. (2002). High-resolution microparticle profiles at NorthGRIP, Greenland: case studies of the calcium–dust relationship. Annals of Glaciology, 35, 237-242.

Kang, J. H., Hwang, H., Hong, S. B., Do Hur, S., Choi, S. D., Lee, J., & Hong, S. (2015). Mineral dust and major ion concentrations in snowpit samples from the NEEM site, Greenland. Atmospheric

Environment, 120, 137-143.

Baldo, C., Formenti, P., Nowak, S., Chevaillier, S., Cazaunau, M., Pangui, E., ... & Shi, Z. (2020). Distinct chemical and mineralogical composition of Icelandic dust compared to northern African and Asian dust. Atmospheric Chemistry and Physics, 20(21), 13521-13539.

McCutcheon, J., Lutz, S., Williamson, C., Cook, J. M., Tedstone, A. J., Vanderstraeten, A., ... & Benning, L. G. (2021). Mineral phosphorus drives glacier algal blooms on the Greenland Ice Sheet. Nature communications, 12(1), 1-11.

Jill Bachelder, Marie Cadieux, Carolyn Liu-Kang, Pérrine Lambert, Alexane Filoche, Juliana Aparecida Galhardi, Madjid Hadioui, Amélie Chaput, Marie-Pierre Bastien-Thibault, Kevin J. Wilkinson, James King & Patrick L. Hayes (2020) Chemical and microphysical properties of wind-blown dust near an actively retreating glacier in Yukon, Canada, Aerosol Science and Technology, 54:1, 2-20, DOI: 10.1080/02786826.2019.1676394

---

## Author Response (AR2)

MS No. acp-2021-94
Title: Differentiation of coarse-mode anthropogenic, marine and dust particles in the high Arctic Islands of Svalbard
Authors: Congbo Song et al.
**Response to reviewers**

Comment on "3.1 Characterization of the k-means derived aerosol types"
In this section, the authors would need to explain that they apportioned the occurrence of coarse-mode aerosols during the study period (mainly from March to October), as described in Lines 6-7.

Response: We added "Note that the relative occurrences of the clusters are limited to the study period (mainly from March to October), and thus with limited coverage of the winter season." In Line 220-221.

Comment on "5. Conclusions"
In this section, the authors had better explain the need for further measurements in winter, because the main conclusions are mostly based on the results from the study period (mainly from March to October). In addition, the authors may need to suggest the possibility that the importance of local sources might still be overlooked, because potentially local sources can not be captured by the backward trajectory analysis, as described in Line 255."

Response: We added "Further measurements in winter season are needed to fill the gaps in the present study. In addition, a more in-depth observation (such as single particle composition) and analysis is needed to better quantify local sources, which might have been overlooked here because local sources are not captured by the backward trajectory analyses." In Line 399-401.